# IP$^2$-RSNN: Bi-level Intrinsic Plasticity Enables Learning-to-learn in Recurrent Spiking Neural Networks

## Abstract

Learning-to-learn (L2L), defined as progressively faster learning across similar tasks, is fundamental to both neuroscience and artificial intelligence. However, its neural basis remains elusive, as most studies emphasize neural population dynamics induced by synaptic plasticity while overlooking adaptations driven by intrinsic neuronal plasticity, which point-neuron models cannot capture. To address the above issue, we develop a recurrent spiking neural network with bi-level intrinsic plasticity (IP$^2$-RSNN). First, based on task demands, a slow meta-intrinsic plasticity determines which intrinsic neuronal properties are learnable, which is preserved throughout subsequent task learning once configured. Second, a fast intrinsic plasticity fine-tunes those learnable properties within each task. Our results indicate that the proposed bi-level intrinsic plasticity plays a critical role in enabling L2L in RSNNs and show that IP$^2$-RSNNs outperform point-neuron recurrent neural networks and self-attention models. Furthermore, our analysis of multi-scale neural dynamics reveals that the bi-level intrinsic plasticity is essential to task-type-specific adaptations at both the neuronal and network levels during L2L, while such adaptations cannot be captured by point-neuron models. Our results suggest that intrinsic plasticity provides significant computational advantages in L2L, shedding light on the design of brain-inspired deep learning models and algorithms.

## 1 Introduction

Learning-to-learn (L2L) exhibits a progressive speedup in learning by distilling prior experience and adapting to novel tasks, and has been widely studied in both neuroscience and artificial intelligence (AI) (Wang et al., 2018; Goudar et al., 2023; Bellec et al., 2018; Finn et al., 2017; Hospedales et al., 2021; Andrychowicz et al., 2016). In AI, L2L research predominantly aims to accelerate learning across tasks by designing explicit meta-objectives (Friedrich & Maziero, 2025; Andrychowicz et al., 2016), meta-optimizers (Ravi & Larochelle, 2017; Finn et al., 2017), or meta-representations (Finn et al., 2018; Wang et al., 2024). In computational neuroscience, L2L studies focus on uncovering mechanisms, employing dynamical systems analysis to investigate how shared neural representations and the reorganization of neural dynamics support flexible computation, thereby providing insight into the mechanisms of L2L at the level of population dynamics (Driscoll et al., 2024; Goudar et al., 2023; Duncker et al., 2020; Dubreuil et al., 2022; Remington et al., 2018). Despite that, our understanding of the neural adaptations induced by intrinsic neuronal plasticity during L2L remains limited.

The existing computational models of L2L are predominantly based on analog recurrent neural networks (RNNs) (Yang et al., 2019; Driscoll et al., 2024; Goudar et al., 2023; Duncker et al., 2020) or on neuromodulated RNNs that introduce modulatory signals to enhance flexibility (Costacurta et al., 2024). All these approaches belong to point-neuron models, with neuromodulation implemented primarily at the level of synaptic plasticity. In contrast, spiking neural networks (SNNs) offer greater biological realism, yet their application to L2L has been limited. Notable examples include LSNNs with adaptive neurons (Bellec et al., 2018) and vanilla RSNNs applied to multi-task learning (Pugavko et al., 2023). However, these SNN models have not fully exploited the rich repertoire of intrinsic plasticity mechanisms of biological neurons.

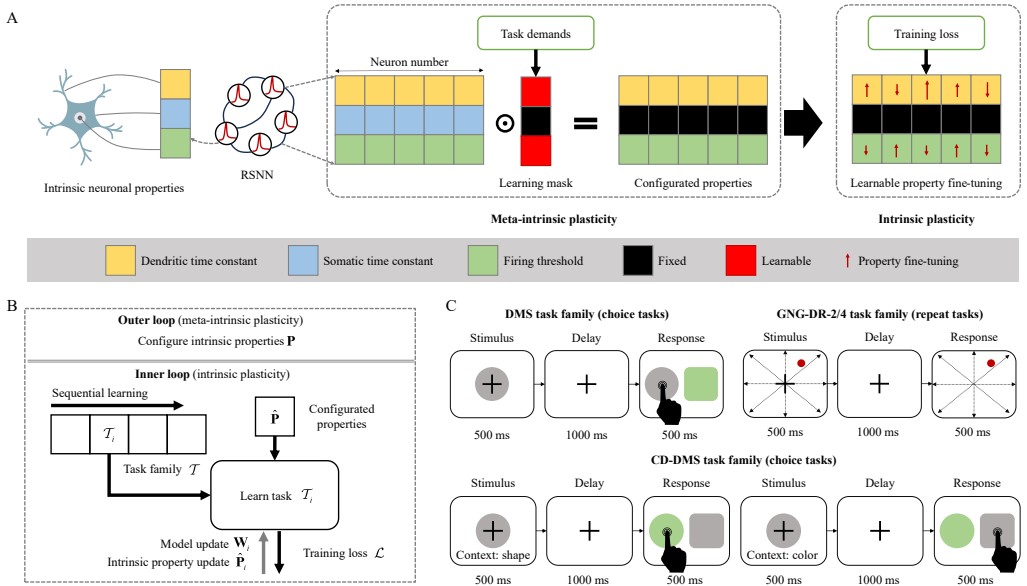

Figure 1: A schematic illustration of the recurrent spiking neural network with bi-level intrinsic plasticity (IP$^2$-RSNN), the learning-to-learn (L2L) paradigm, and the task families. (A) IP$^2$-RSNN incorporates meta-intrinsic plasticity and intrinsic plasticity to regulate intrinsic neuronal properties. Three intrinsic properties are considered in our work: dendritic time constant, somatic time constant, and firing threshold. (B) The L2L paradigm consists of two loops: the outer loop implements meta-intrinsic plasticity, and the inner loop applies intrinsic plasticity during the sequential learning of tasks from the task family $\mathcal{T}$. (C) We evaluate four task families: delayed match-to-sample (DMS), context-dependent DMS (CD-DMS), and two go/no-go delayed recall tasks (GNG-DR-2, GNG-DR-4). These tasks fall into two categories: choice tasks (DMS, CD-DMS), which require stimulus–decision mapping; and repeat tasks (GNG-DR-2, GNG-DR-4), which require delayed signal recall.

To address these challenges, this work makes the following contributions.

- We introduce a recurrent spiking neural network with bi-level intrinsic plasticity (IP$^2$-RSNN). This model combines a slow meta-intrinsic plasticity mechanism, which determines the learnability of different intrinsic neuronal properties and preserves this configuration across sequential task learning, with a fast intrinsic plasticity mechanism that fine-tunes those learnable properties within each task. By leveraging this bi-level intrinsic plasticity, we are able to investigate the neural adaptations induced by intrinsic neuronal plasticity.

- We demonstrate the necessity of bi-level intrinsic plasticity for enabling L2L in RSNNs. Moreover, IP$^2$-RSNNs achieve superior performance compared to four point-neuron artificial neural networks (ANNs), namely one RNN and three simplified self-attention models.

- Our analysis of multi-scale neural dynamics reveals that bi-level intrinsic plasticity is a key factor enabling RSNNs to adapt in a task-type-specific manner at both the neuronal and network levels during L2L. Such adaptation does not emerge in point-neuron ANNs.

## 2 MODEL

### 2.1 IP$^2$-RSNN

Our IP$^2$-RSNN includes three intrinsic properties (Fig. 2A): dendritic time constants $\mathbf{P}_{\tau_d}$, which regulate dendritic integration by adaptively weighting short-term and long-term signals (Zheng et al., 2024; Branco & Häusser, 2010; Magee, 2000; Zenke et al., 2017); somatic time constants $\mathbf{P}_{\tau_s}$,

which support memory maintenance by adjusting the decay rate of the membrane potential (Zheng et al., 2024; Hasselmo, 2006); and firing thresholds $\mathbf{P}_\theta$, which support complex tasks by controlling neuronal firing to form complex decision boundaries (Bellec et al., 2018; Daoudal & Debanne, 2003; Titley et al., 2017; Triesch, 2004). The detailed neuronal model can be found in the Appendix A.1.

The meta-intrinsic plasticity in IP$^2$-RSNN configures the learnability of these properties (Fig. 2A):

$$\hat{\mathbf{P}} = \mathbf{m} \odot \mathbf{P}^{\text{learnable}} + (\mathbf{1} - \mathbf{m}) \odot \mathbf{P}^{\text{fixed}}, \tag{1}$$

$$\hat{\mathbf{P}}_{\tau_d} = m_1 \cdot \mathbf{P}_{\tau_d}^{\text{learnable}} + (1 - m_1) \cdot \mathbf{P}_{\tau_d}^{\text{fixed}},$$
$$\hat{\mathbf{P}}_{\tau_s} = m_2 \cdot \mathbf{P}_{\tau_s}^{\text{learnable}} + (1 - m_2) \cdot \mathbf{P}_{\tau_s}^{\text{fixed}}, \tag{2}$$
$$\hat{\mathbf{P}}_\theta = m_3 \cdot \mathbf{P}_\theta^{\text{learnable}} + (1 - m_3) \cdot \mathbf{P}_\theta^{\text{fixed}},$$

where $\mathbf{m}(\mathcal{T}) = [m_1 \quad m_2 \quad m_3]$ is the learning mask based on the demands of the task family $\mathcal{T}$; $m_1 \in \{0, 1\}$ indicates whether dynamic integration is required in $\mathcal{T}$, $m_2 \in \{0, 1\}$ specifies the need for high-fidelity temporal memory, and $m_3 \in \{0, 1\}$ denotes the demand for complex decision-making. $\hat{\mathbf{P}} = \{\hat{\mathbf{P}}_{\tau_d}, \hat{\mathbf{P}}_{\tau_s}, \hat{\mathbf{P}}_\theta\}$ denotes the configured properties, and $\mathbf{P}^{\text{learnable}} = \{\mathbf{P}_{\tau_d}^{\text{learnable}}, \mathbf{P}_{\tau_s}^{\text{learnable}}, \mathbf{P}_\theta^{\text{learnable}}\}$ and $\mathbf{P}^{\text{fixed}} = \{\mathbf{P}_{\tau_d}^{\text{fixed}}, \mathbf{P}_{\tau_s}^{\text{fixed}}, \mathbf{P}_\theta^{\text{fixed}}\}$ are the candidate properties.

The intrinsic plasticity in IP$^2$-RSNN fine-tunes the learnable properties during task learning (Fig. 2A):

$$\hat{\mathbf{P}}_i \leftarrow \hat{\mathbf{P}}_i - \eta \odot \nabla_{\hat{\mathbf{P}}_i} \mathcal{L}, \tag{3}$$

where $\eta$ is the learning rate and $\mathcal{L}$ is the task-specific loss function. For fixed properties, their gradients are zero, so only learnable properties are updated. The task loss function $\mathcal{L}$ comprises four components:

$$\mathcal{L} = \mathcal{L}_{\text{Base}} + \lambda_h \mathcal{L}_h + \lambda_{\text{in}} \mathcal{L}_{\text{in}} + \lambda_{\text{rec}} \mathcal{L}_{\text{rec}} + \lambda_{\text{out}} \mathcal{L}_{\text{out}}, \tag{4}$$

$$\mathcal{L}_h = \left| \frac{1}{H} \sum_{h=1}^{H} h_h^2 - \sigma_h^2 \right|, \quad \mathcal{L}_{\text{in}} = \frac{1}{|\mathbf{W}_{\text{in}}|} \sum_{i,j} \mathbf{W}_{\text{in},ij}^2,$$

$$\mathcal{L}_{\text{rec}} = \frac{1}{|\mathbf{W}_{\text{rec}}|} \sum_{i,j} \mathbf{W}_{\text{rec},ij}^2, \quad \mathcal{L}_{\text{out}} = \frac{1}{|\mathbf{W}_{\text{out}}|} \sum_{i,j} \mathbf{W}_{\text{out},ij}^2, \tag{5}$$

where $\mathcal{L}_{\text{Base}}$ is the base term measuring the discrepancy between the ground truth and the model prediction; $\mathbf{W}_{\text{in}}$, $\mathbf{W}_{\text{rec}}$, and $\mathbf{W}_{\text{out}}$ are the input, recurrent, and output weight matrices (their corresponding regularization terms are $\mathcal{L}_{\text{in}}$, $\mathcal{L}_{\text{rec}}$, and $\mathcal{L}_{\text{out}}$); and $\mathcal{L}_h$ is a homeostatic term on hidden-state dynamics to stabilize learning. Here, $\lambda_h$, $\lambda_{\text{in}}$, $\lambda_{\text{rec}}$, and $\lambda_{\text{out}}$ are hyperparameters controlling the weight of each loss term. $H$ denotes the number of hidden units; $h_h$ denotes the activation of the hidden unit $h$; and $\sigma_h^2$ is the target mean squared activation, initially set to zero. When the model transitions to a new task, it is updated based on the hidden-unit activity from the previous task.

## 2.2 L2L PARADIGM AND TASKS

The L2L paradigm includes two loops (Fig. 2B): the outer loop implements meta-intrinsic plasticity (Eq. 1) and produces the configured properties $\hat{\mathbf{P}}$. This configuration is preserved in the inner loop; the inner loop implements intrinsic plasticity (Eq. 3) and updates the model during the sequential learning of tasks. It serves to verify whether bi-level intrinsic plasticity enables learning-to-learn to arise purely from the natural dynamics of learning. Updates to $\hat{\mathbf{P}}_i$ and the model parameters $\mathbf{W}_i$ are driven by the training loss $\mathcal{L}$ for task $\mathcal{T}_i$ from the task family $\mathcal{T}$ (Eq. 4). We set the number of tasks within each task family to 1000 in order to fully observe the speedup trend and the neural dynamics during L2L.

To comprehensively characterize and compare neural dynamics during L2L, we adopt four task families (Fig. 2C; Appendix A.2) from neuroscience experiments: delayed match-to-sample (DMS) (Britten et al., 1992), context-dependent delayed match-to-sample (CD-DMS) (Mante et al., 2013), and two go/no-go delayed recall task sets (GNG-DR-2 and GNG-DR-4) (Funahashi et al., 1989; Mendoza-Halliday & Martinez-Trujillo, 2017). Tasks in these families all comprise three consecutive periods: stimulus (500 ms), delay (1000 ms), and response (500 ms). DMS and CD-DMS require mapping high-dimensional stimuli to binary decisions after a delay, with CD-DMS adding contextual complexity. GNG-DR-2 and GNG-DR-4 require stimulus reproduction after a delay, with

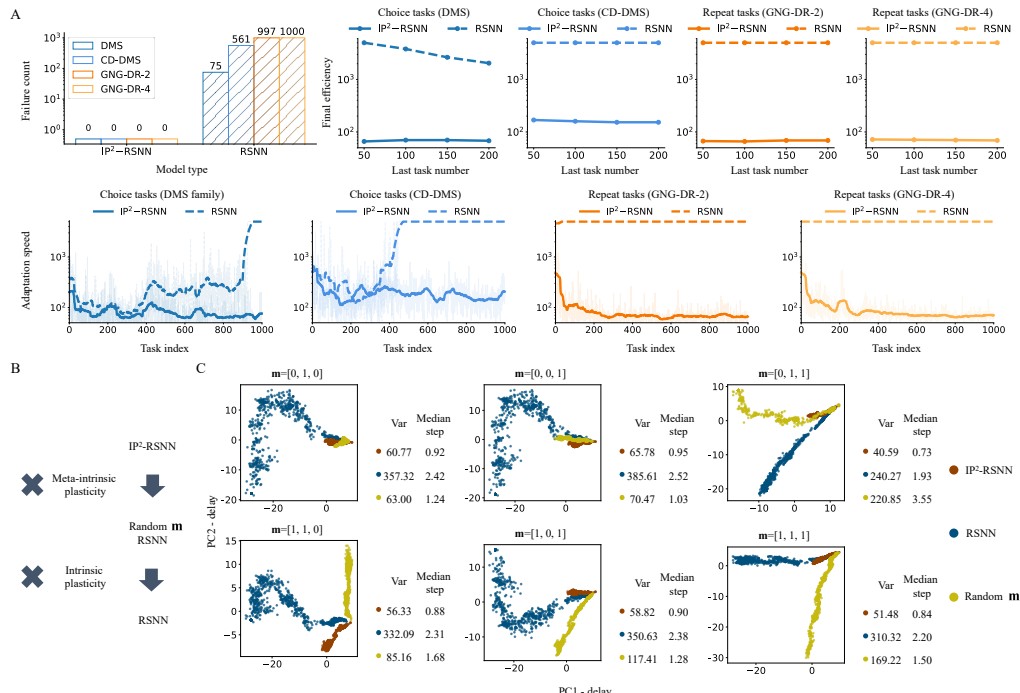

Figure 2: Effects of bi-level intrinsic plasticity and disentangled role of the meta-intrinsic plasticity and intrinsic plasticity. (A) L2L performance comparison between IP$^2$-RSNN and RSNN across four task families, evaluated by three quantitative indicators: failure count, adaptation speed, and final efficiency. (B) Roles of meta-intrinsic plasticity and intrinsic plasticity in IP$^2$-RSNN disentangled by randomizing the learning mask $\mathbf{m}$. (C) Comparison of low-dimensional representations among IP$^2$-RSNN, vanilla RSNN, and RSNN variants with randomized $\mathbf{m}$, obtained from the delay period across all DMS tasks using principal component analysis (PCA).

GNG-DR-4 further increasing input dimensionality relative to GNG-DR-2. Based on the underlying task rules, the four task families fall into two categories: choice tasks (decision making) and repeat tasks (reproduction).

All tasks above involve multidimensional temporal stimuli, thereby requiring dynamic input integration ($m_1 = 1$ in all cases). Compared to DMS and GNG-DR-2, CD-DMS and GNG-DR-4 place higher demands on decision-making complexity ($m_3 = 1$), whereas GNG-DR-2 and GNG-DR-4 additionally require higher temporal fidelity of memory ($m_2 = 1$). Thus, the learning masks and the corresponding configured properties of these four task families are specified as follows:

$$\mathbf{m}(\text{DMS}) = [1, 0, 0], \quad \mathbf{m}(\text{CD-DMS}) = [1, 0, 1],$$
$$\mathbf{m}(\text{GNG-DR-2}) = [1, 1, 0], \quad \mathbf{m}(\text{GNG-DR-4}) = [1, 1, 1]. \tag{6}$$

$$\hat{\mathbf{P}}_{\text{DMS}} = \{\mathbf{P}_{\tau_d}^{\text{learnable}}, \mathbf{P}_{\tau_s}^{\text{fixed}}, \mathbf{P}_{\theta}^{\text{fixed}}\}, \quad \hat{\mathbf{P}}_{\text{CD-DMS}} = \{\mathbf{P}_{\tau_d}^{\text{learnable}}, \mathbf{P}_{\tau_s}^{\text{fixed}}, \mathbf{P}_{\theta}^{\text{learnable}}\},$$
$$\hat{\mathbf{P}}_{\text{GNG-DR-2}} = \{\mathbf{P}_{\tau_d}^{\text{learnable}}, \mathbf{P}_{\tau_s}^{\text{learnable}}, \mathbf{P}_{\theta}^{\text{fixed}}\}, \quad \hat{\mathbf{P}}_{\text{GNG-DR-4}} = \{\mathbf{P}_{\tau_d}^{\text{learnable}}, \mathbf{P}_{\tau_s}^{\text{learnable}}, \mathbf{P}_{\theta}^{\text{learnable}}\}. \tag{7}$$

## 3 RESULTS

### 3.1 BI-LEVEL INTRINSIC PLASTICITY IS ESSENTIAL FOR LEARNING-TO-LEARN

To examine whether bi-level intrinsic plasticity facilitates learning-to-learn, we compare the proposed IP$^2$-RSNN with a standard RSNN across four task families (Fig. 2). We evaluate L2L performance using three metrics: (i) failure count, the number of tasks within a family on which the model fails to converge; (ii) adaptation speed, the number of iterations required for task $\mathcal{T}_i$ to reach a predefined convergence threshold; and (iii) final efficiency, the average adaptation speed over the last 50,

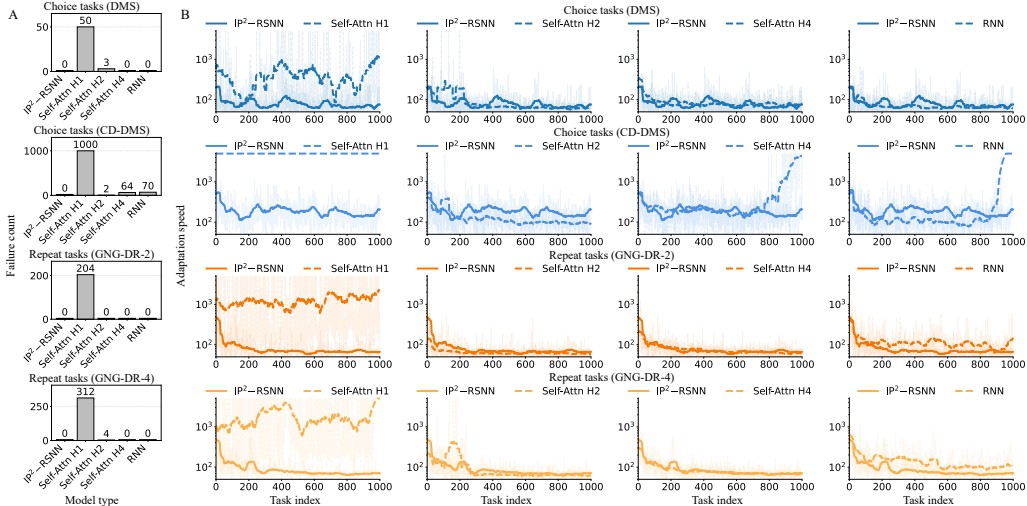

Figure 3: L2L performance comparison between IP$^2$-RSNN and four artificial neural networks (ANNs), including a standard recurrent neural network (RNN) and three Transformer-based models (Vaswani et al., 2017), with one, two, or four attention heads (Self-Attn H1, H2, H4). (A) Comparison of failure counts across four task families. (B) Comparison of final efficiency across four task families.

100, 150, and 200 tasks. IP$^2$-RSNN incurs no failures in any task family, whereas the RSNN exhibits substantial failures (Fig. 2A). For adaptation speed, IP$^2$-RSNN decreases consistently with the task index across all families, indicating that the network acquires L2L during sequential exposure; by contrast, the RSNN shows no such a decreasing trend (Fig. 2A). Finally, IP$^2$-RSNN achieves much better final efficiency than the RSNN (Fig. 2A). These results demonstrate that bi-level intrinsic plasticity is essential for enabling learning-to-learn in RSNNs.

To test whether both levels of intrinsic plasticity are necessary, we conduct ablation studies and disentangle their roles by randomizing the learning mask $\mathbf{m}$ (Fig. 2B). We probe the contribution of meta-intrinsic plasticity by comparing low-dimensional delay-period activity across all DMS tasks between IP$^2$-RSNN and RSNNs with randomized $\mathbf{m}$. Randomizing $\mathbf{m}$ markedly increases both the variance and the median step size, indicating that meta-intrinsic plasticity enhances representational consistency and thereby facilitates L2L (Fig. 2C). To isolate the contribution of intrinsic plasticity, we compare the vanilla RSNN with RSNNs using randomized $\mathbf{m}$. Compared to the vanilla RSNN, most random masks yield a lower variance and a smaller median step size; however, for the specific mask $\mathbf{m} = [0, 1, 1]$, the performance degrades. This suggests that intrinsic plasticity improves L2L by providing an additional tuning pathway beyond synaptic plasticity. Note, however, that amplifying plasticity along irrelevant dimensions can be detrimental (Fig. 2C). Overall, IP$^2$-RSNN produces the most compact state space, followed by most RSNNs with random $\mathbf{m}$, then standard RSNNs, with RSNNs using maladaptive masks performing worst (Fig. 2C). The L2L performance on four task families further supports this ranking (Fig. S1). These results indicate that meta-intrinsic plasticity and intrinsic plasticity act synergistically to regulate intrinsic properties and enable learning-to-learn.

### 3.2 IP$^2$-RSNN ACHIEVES SUPERIOR PERFORMANCE COMPARED WITH ANNS

We compare IP$^2$-RSNN with a set of ANNs, including a standard RNN and three Transformer-based models (Vaswani et al., 2017) with one, two, or four attention heads (Self-Attn H1, H2, H4), to demonstrate the competitive performance of IP$^2$-RSNN in L2L. Notably, none of the ANN models are able to successfully complete all sequential tasks across the four task families, whereas IP$^2$-RSNN achieves full task completion (Fig. 3A). Moreover, in the DMS, GNG-DR-2, and GNG-DR-4 task families, IP$^2$-RSNN exhibits a comparable adaptation speed to that of the best-performing ANN models (Fig. 3B). In the CD-DMS task family, the adaptation trend of IP$^2$-RSNN is slightly worse than Self-Attn H2; however, the IP$^2$-RSNN completes all 1000 sequential tasks, whereas Self-

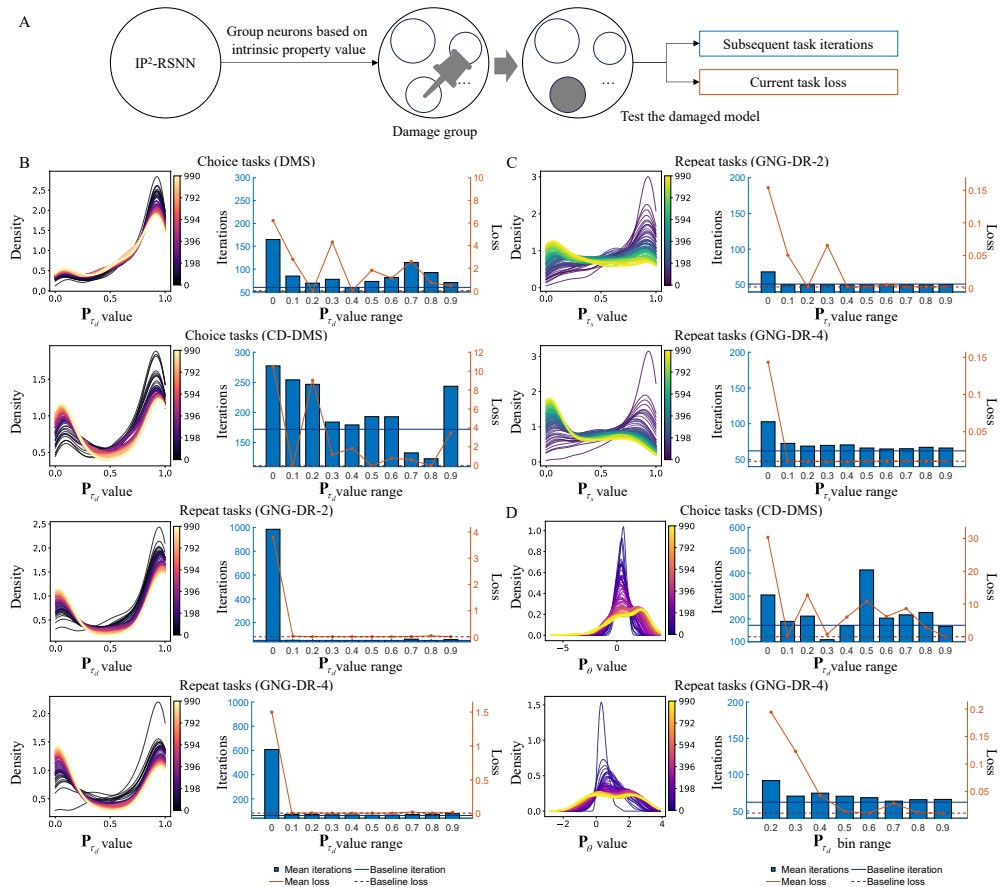

Figure 4: Analysis of neuron-level adaptation in IP²-RSNN. (A) Method to capture functional roles of neurons. Neurons are first grouped based on their intrinsic property values, and functional roles are probed by silencing neurons within each group. The impact is assessed by examining subsequent-task learning iterations and current-task execution loss of the damaged IP²-RSNN. (B) Evolution of $\mathbf{P}_{\tau_d}$ during L2L in four task families, and effects of damage on next-task iterations and current-task loss for neuron groups stratified by $\mathbf{P}_{\tau_d}$ ranges. (C) Evolution of $\mathbf{P}_{\tau_s}$ during L2L in the two repeat task families, and effects of damage on next-task iterations and current-task loss for neuron groups stratified by $\mathbf{P}_{\tau_s}$ ranges. (D) Evolution of $\mathbf{P}_{\theta}$ during L2L in CD-DMS and GNG-DR-4, and effects of damage on next-task iterations and current-task loss for neuron groups stratified by $\mathbf{P}_{\theta}$ ranges.

Attn H2 fails on two (Fig. 3A). These results indicate that IP²-RSNN achieves better performance than ANN models.

## 3.3 NEURONAL-LEVEL ADAPTATION IN IP²-RSNN

Based on bi-level intrinsic plasticity, the intrinsic properties of IP²-RSNN can evolve dynamically during L2L. Thus, we group neurons based on their intrinsic property values and examine the group functional roles by damaging the corresponding group in IP²-RSNN and testing the damaged IP²-RSNN's subsequent-task learning iterations and current-task execution loss (Fig. 4A). In this work, intrinsic property values of IP²-RSNN are extracted from the penultimate training task (task index = 999) in each task family, because the intrinsic property evolution trend is stable on this task. Moreover, intrinsic property values are normalized and divided into ten equal ranges. Neurons falling in the same intrinsic property range represent a neuronal group. This design parallels lesion analysis in biological systems, enabling inference of functional roles for distinct neuronal subgroups (Vaidya et al., 2019).

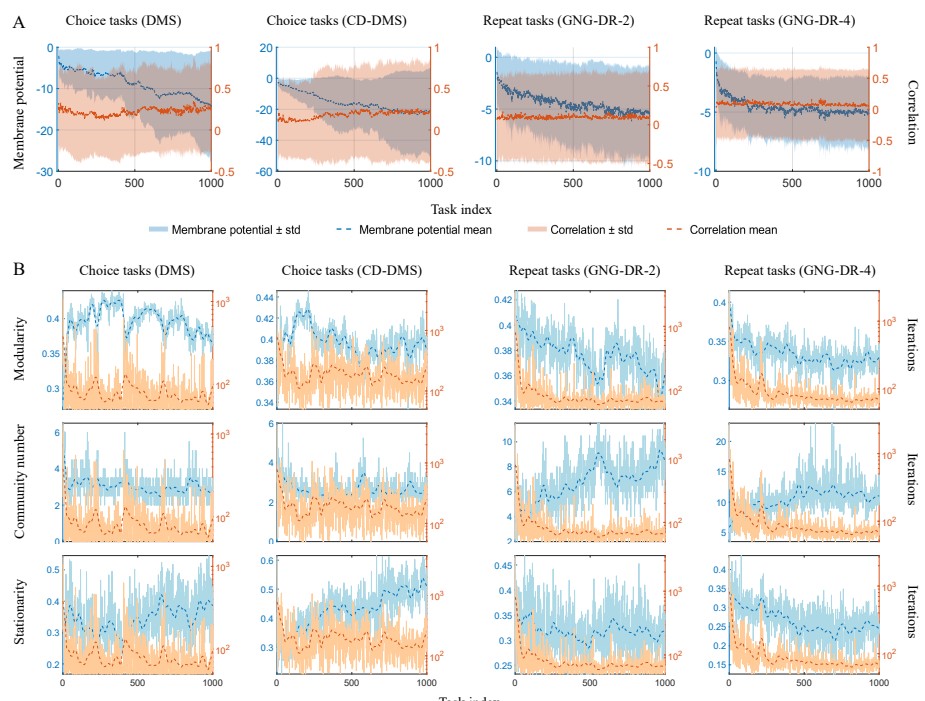

Figure 5: Analysis of the network level adaptation in IP²-RSNN across four task families. (A) Dynamics of membrane potentials and their correlations. Membrane potentials are examined because they capture both subthreshold fluctuations and near-threshold states that are not reflected in spikes. (B) Trends in modularity, community count, and stationarity plotted against iterations.

We first examine the evolution of neuronal intrinsic properties during L2L (Fig. 4B). For dendritic time constants $\mathbf{P}_{\tau_d}$, we observe a consistent shift in the distribution across all four task families: initially concentrated near 1, the distribution gradually becomes bimodal, with emerging peaks near 0 and 1. Furthermore, task-family comparisons reveal that $\mathbf{P}_{\tau_d,\text{CD-DMS}}$ and $\mathbf{P}_{\tau_d,\text{GNG-DR-4}}$ exhibit stronger shifts toward shorter time constants than $\mathbf{P}_{\tau_d,\text{DMS}}$ and $\mathbf{P}_{\tau_d,\text{GNG-DR-2}}$, suggesting that more complex task inputs drive more extreme reductions in dendritic integration windows. For somatic time constants $\mathbf{P}_{\tau_s}$, we find a similar but slightly more pronounced trend in the GNG-DR-2 and GNG-DR-4 task families: the dominant peak shifts from around 1 toward 0. This indicates that neurons progressively adopt shorter somatic integration times, enabling fast, high-resolution processing that is critical for accurate memory recall. Consistent with the dendritic parameters, tasks with a heavier memory burden also induce stronger reductions in $\mathbf{P}_{\tau_s}$, reinforcing the role of somatic time constants. For firing thresholds $\mathbf{P}_\theta$, the overall distribution gradually broadens and shifts toward higher values. This adjustment facilitates threshold diversity, contributing to the refinement of decision boundaries.

We then analyze damage effects (Fig. 4B) across different intrinsic properties. For $\mathbf{P}_{\tau_d}$ derived damage, both the next-task iterations and the current-task loss exhibit prominent peaks in the 0-0.1 range across all four task families, indicating that neuron groups with small $\mathbf{P}_{\tau_d}$ values play critical roles in both task execution and L2L capability. Comparing repeat tasks (GNG-DR-2 and GNG-DR-4) with choice tasks (DMS and CD-DMS), we observe that neurons in repeat tasks exhibit an extremely clear functional division: only two functional roles emerge, including a dominant mixed group in the 0-0.1 range and a redundant group elsewhere. By contrast, choice tasks show more complex functional differentiation. For $\mathbf{P}_{\tau_s}$ derived damage, the influence on current-task loss is stronger than on next-task iterations, indicating that $\mathbf{P}_{\tau_s}$ based groups mainly contribute to task processing. Consistent with $\mathbf{P}_{\tau_d}$, repeat tasks still exhibit a clear functional segregation. Finally, for $\mathbf{P}_\theta$ derived damage, current-task loss peaks in smaller bin ranges across the CD-DMS and GNG-DR-4 task families. In CD-DMS, next-task iterations peak around the 0.5-0.6 range. By contrast, in GNG-DR-4, firing thresholds have limited influence on next-task iterations.

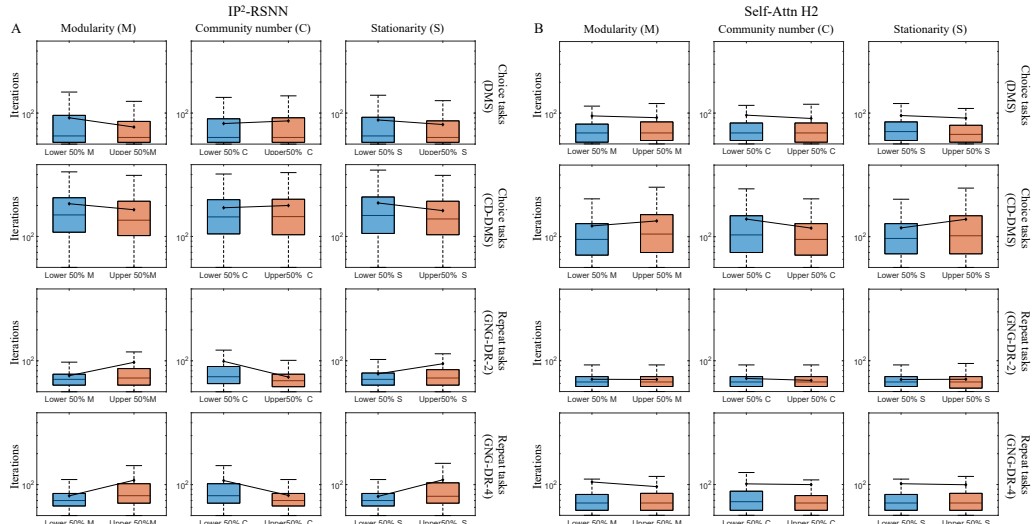

Figure 6: Relationship between IP$^2$-RSNN modularity and L2L adaptation speed. (A) Relationships between three modularity metrics (modularity, number of communities, and stationarity) and adaptation speed in IP$^2$-RSNN across four task families. For each metric, the 1000 tasks are sorted in ascending order of their metric values and split at the median into lower 50% and upper 50% groups. Thus adaptation speed distributions can be compared between the two groups. (B) Relationships between three modularity metrics (modularity, number of communities, and stationarity) and adaptation speed in Self-Attn H2 across four task families. Self-Attn H2 is shown because it performs best among the three attention-based ANN models.

The damage results show that task families in the same category exhibit more similar functional role patterns in IP$^2$-RSNN.

### 3.4 NETWORK-LEVEL ADAPTATION IN IP$^2$-RSNN

We first investigate the dynamics, correlations, and distributional evolution of membrane potentials during L2L in IP$^2$-RSNN (Fig. 5A, Fig. S2). Across all task families, the mean membrane potential gradually decreases, while its variance steadily increases, indicating that overall activation patterns become sparser and more differentiated (Fig. 5A). For membrane potential correlations, the mean values remain positive and relatively stable, while the correlation variance is large, suggesting that some neural subpopulations work in coordination whereas others operate independently (Fig. 5A). Compared with repeat tasks (GNG-DR-2 and GNG-DR-4), choice tasks (DMS and CD-DMS) show more negative mean potentials but higher mean correlations, reaching around 0.25, whereas repeat tasks remain only slightly above zero (Fig. 5A). These results reflect task-type-specific adaptation of membrane potentials in IP$^2$-RSNN.

Neurophysiological studies suggest that functional modularity is a hallmark of flexible and efficient neural computation, enabling the brain to reorganize subnetworks to meet task demands (Bassett & Sporns, 2017; Shine et al., 2016). We therefore investigate the modularity dynamics of IP$^2$-RSNN during L2L. We first examine the trends of modularity, the number of communities, and stationarity (Fig. 5B) and visualize the evolution of the modular allegiance matrix during L2L across the four task families (Fig. S3). In the choice tasks (DMS and CD-DMS), modularity first increases with fluctuations and then gradually declines, indicating that the network initially forms distinct modules to meet task demands but later dissolves boundaries to enhance integration and support L2L. Consistently, the number of communities decreases while stationarity increases, suggesting that modules are reused and their functional roles become more stable. By contrast, in the repeat tasks (GNG-DR-2 and GNG-DR-4), modularity decreases, the number of communities increases, and stationarity declines. This reorganization reflects the demands of high-fidelity memory, requiring detailed information to be distributed across multiple subpopulations. Accordingly, the network weakens modular

boundaries to improve inter-community coordination, forms more small specialized communities, and maintains flexible, dynamically reconfigurable functional roles.

We then analyze the relationship between three modularity metrics and L2L adaptation speed. For each metric, we sort the 1000 tasks in ascending order of their values and split them at the median into lower 50% and upper 50% groups. We then compare adaptation speed distributions between the two groups to assess how each modularity metric relates to L2L efficiency. In the choice tasks (DMS and CD-DMS), modularity is negatively correlated with the number of iterations required, indicating that stronger modular organization facilitates more efficient L2L (Fig. 6A). By contrast, the number of communities is positively correlated with the number of iterations required, suggesting that overly fine-grained community partitioning may hinder L2L capability (Fig. 6A). Stationarity is also negatively correlated with the number of iterations, implying that more stable functional roles of modules support improved learning efficiency (Fig. 6A). By contrast, these relationships reverse in the repeat tasks (GNG-DR-2 and GNG-DR-4): higher modularity, lower community number, and higher stationarity are all associated with more iterations (Fig. 6A). These results show that modularity of IP$^2$-RSNN adapts in a task-type-specific manner in L2L. Notably, ANNs such as RNNs and Self-Attn H2 cannot capture this (Fig. 6B, Fig. S4).

## 4 RELATED WORK

A growing body of behavioral, fMRI, and computational work suggests that the prefrontal cortex (PFC) can function as a meta-reinforcement learner. Under dopaminergic training signals, the PFC acquires an internal learning loop that supports rapid cross-task adaptation, i.e., learning to learn (Wang et al., 2018; Eichenbaum et al., 2020; Stokes et al., 2013; Miller & Cohen, 2001). Causal evidence also implicates the orbitofrontal cortex (OFC) in meta-learning. In mice, reversal learning generalizes across contexts with progressively faster reinforcement learning, and disrupting OFC function, either via inhibition or by blocking CaMKII-dependent synaptic plasticity, abolishes this L2L effect (Hattori et al., 2023). In addition, the PFC together with the hippocampus (HPC) and medial temporal lobe (MTL) supports abstract, task-agnostic strategies and relational representations that provide a substrate for rapid transfer and flexible generalization across contexts (Behrens et al., 2018; Stachenfeld et al., 2017). Despite this evidence, the neural mechanisms underlying L2L at the microscopic level of intrinsic and synaptic plasticity remain unclear, owing to the difficulty of performing multi-scale biological experiments. Computational modeling therefore offers a powerful framework to probe these mechanisms.

Computational studies of L2L mechanisms have largely focused on how circuits reconfigure across tasks and reuse low-dimensional subspaces and population dynamics. However, these studies typically examine flexibility under task-specific training objectives rather than directly demonstrating progressive speedup without an explicit meta-objective (Driscoll et al., 2024; Goudar et al., 2023; Duncker et al., 2020; Dubreuil et al., 2022; Remington et al., 2018). Work explicitly addressing such meta-objective-free progressive speedup is limited. The first demonstration, to our knowledge, is Goudar et al. (2023), which showed that during L2L an RNN can form and reuse low-dimensional manifolds to accelerate learning, but this analysis is confined to the population level and does not probe cellular mechanisms. In this study, we show that standard training in IP$^2$-RSNNs naturally gives rise to L2L, and we provide a cellular-level account of the phenomenon, thereby filling the gap between circuit-level dynamics and neuron-level mechanisms.

## 5 CONCLUSION

We propose IP$^2$-RSNN, which integrates bi-level intrinsic plasticity: meta-intrinsic plasticity configures the learnability of intrinsic neuronal properties, while intrinsic plasticity fine-tunes those properties during task learning. We show that bi-level intrinsic plasticity is essential for L2L in RSNNs and that its two components act synergistically to regulate intrinsic properties. We further find that IP$^2$-RSNNs outperform point-neuron ANNs. Analyses of multi-scale neural dynamics further reveal that bi-level intrinsic plasticity enables RSNNs to adapt in a task-type-specific manner at both the neuronal and network levels during L2L. Such adaptations do not emerge in point-neuron ANNs. These findings highlight the computational advantages of brain-inspired models and inform the design of more efficient neural architectures.

# 6 ETHICS STATEMENT

This submission does not present any ethical concerns.

# 7 REPRODUCIBILITY STATEMENT

We will upload a source code ZIP in the supplementary material. A public GitHub repository will be released upon acceptance.

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

# A  APPENDIX A

## A.1  DETAILED NEURAL MODEL IN IP$^2$-RSNN

The membrane potential dynamics of spiking neurons in IP$^2$-RSNNs are defined as:

$$V(t) = \mathbf{P}_{\tau_s}V(t-1) + (1 - \mathbf{P}_{\tau_s})\Big(\mathbf{W}_{\text{in}}x_{\text{in}}(t) + \mathbf{W}_{\text{rec}}S_{\text{mem}}(t-1) + N(t)\Big), \tag{8}$$

where $\mathbf{W}_{\text{in}}$ and $\mathbf{W}_{\text{rec}}$ are the input and recurrent weight matrices, $\mathbf{P}_{\tau_s}$ is the decay factor parameter associated with the somatic time constant, $x_{\text{in}}(t)$ denotes the input at time step $t$, $S_{\text{mem}}(t-1)$ is the membrane state from the previous step, and $N(t)$ is a noise term updated as:

$$N(t+1) = (1 - \alpha_{\text{noise}})N(t) + \sqrt{2\alpha_{\text{noise}}}\, A_{\text{noise}}\, \mathcal{N}(0,1), \tag{9}$$

with $\alpha_{\text{noise}}$ the noise decay constant, $A_{\text{noise}}$ the noise amplitude, and $\mathcal{N}(0,1)$ standard Gaussian noise. When dendritic integration is taken into account, the membrane potential then evolves as:

$$V(t) = \mathbf{P}_{\tau_s}V(t-1) + (1 - \mathbf{P}_{\tau_s})$$
$$\left( \sum_d \mathbf{P}_{\tau_d,d}V_d(t-1) + (1 - \mathbf{P}_{\tau_d,d})(\mathbf{W}_{\text{in},d}x_{\text{in}}(t) + \mathbf{W}_{\text{rec},d}S_{\text{mem}}(t-1)) \right) + N(t), \tag{10}$$

where $d$ indexes dendritic branches, and $\mathbf{P}_{\tau_d,d}$ is the decay factor parameter of dendritic compartment $d$. A spike is emitted when the membrane potential crosses a threshold $\mathbf{P}_\theta$:

$$S_{\text{spike}}(t) = \begin{cases} 1, & \text{if } V(t) \geq \mathbf{P}_\theta, \\ 0, & \text{otherwise.} \end{cases} \tag{11}$$

Upon spiking, the membrane potential is reset to $V_{\text{reset}}$. Membrane state evolves as:

$$S_{\text{mem}}(t) = \alpha S_{\text{mem}}(t-1) + (1 - \alpha)S_{\text{spike}}(t-1), \tag{12}$$

where $\alpha$ controls the temporal smoothing of spiking activity. The network output is given by:

$$\hat{Y}(t) = \mathbf{W}_{\text{out}}S_{\text{mem}}(t), \tag{13}$$

with $\mathbf{W}_{\text{out}}$ denoting output weights.

## A.2  DETAILED TASK DESCRIPTION

We consider four task families. The DMS family (Britten et al., 1992) follows the configuration in Goudar et al. (2023). Each task comprises two input trials, $x_{\text{in},1}$ and $x_{\text{in},2}$, both spanning three periods: stimulus (500 ms), delay (1000 ms), and response (500 ms). During the stimulus period, the input contains a 10-dimensional stimulus signal and a 1-dimensional fixation signal. During the delay period, only the fixation signal is maintained. During the response period, both signals are set to zero. The corresponding target trials, $y_1$ and $y_2$, preserve the fixation signal during stimulus and delay, and switch to a categorical label encoding stimulus identity in the response period. The CD-DMS family (Mante et al., 2013) extends the DMS family by introducing a binary context cue during the stimulus period. When the cue is 0, the stimulus-response mapping is identical to the DMS; when the cue is 1, the mapping is reversed. Each task therefore contains four input-target trial pairs. Two GNG-DR families (Funahashi et al., 1989; Mendoza-Halliday & Martinez-Trujillo, 2017) require the model to either reproduce the input stimulus (go) or suppress the output near zero (no-go) during the response period. Two variants are used: GNG-DR-2 with two-dimensional inputs and GNG-DR-4 with four-dimensional inputs. Each task in the task families has input stimuli that are randomly generated and different.

Based on underlying task rules, the four task families fall into two categories. The choice tasks (DMS and CD-DMS) require mapping high-dimensional stimuli to binary decisions, with CD-DMS adding contextual complexity. The repeat tasks (GNG-DR-2 and GNG-DR-4) require stimulus reproduction after a delay, with GNG-DR-4 further increasing input dimensionality relative to GNG-DR-2.

Table 1: Parameter configuration in this study. RSNNs in this table include IP²-RSNN, vanilla RSNN, and RSNNs with randomized $\mathbf{m}$. Moreover, $\beta_1$ and $\beta_2$ denote the decay rates of the first and second moments in the Adam optimizer (Adam et al., 2014).

| Task and L2L Configuration | | | | |
|---|---|---|---|---|
| | DMS | CD-DMS | GNG-DR-2 | GNG-DR-4 |
| Sample/Fixation Dimension | 10/1 | 11/1 | 2/1 | 4/1 |
| Response/Fixation Dimension | 2/1 | 2/1 | 2/1 | 4/1 |
| Stimulus/Delay/Response Duration (ms) | | | 500/1000/500 | |
| Task Number of Each Family | | | 1000 | |
| Maximum Iterations per Task | | | 5000 | |
| Minimum Iterations per Task | | | 50 | |
| Consecutive Failure for L2L Early Stopping | | | 3 | |
| Convergence Loss Threshold | | 0.005 (DMS, CD-DMS, GNG-DR-2) / 0.006 (GNG-DR-4) | | |
| **Training Configuration** | | | | |
| Optimizer | | | Adam | |
| Learning Rate (All ANNs/All RSNNs) | | | 0.0001 / 0.01 | |
| $\beta_1$ (All ANNs/All RSNNs) | | | 0.3 / 0.1 | |
| $\beta_2$ (All ANNs/All RSNNs) | | | 0.999 / 0.3 | |
| $\mathcal{L}_{\text{Base}}$ | | CE Loss (DMS, CD-DMS) / MSE Loss (GNG-DR-2, GNG-DR-4) | | |
| $\lambda_h$ (All ANNs/All RSNNs) | | | 0.0005 / 0.0005 | |
| $\lambda_{\text{in}}$ (RNN/All RSNNs) | | | 0.001 / 0.001 | |
| $\lambda_{\text{rec}}$ (RNN/All RSNNs) | | | 0.0001 / 0.0001 | |
| $\lambda_{\text{out}}$ (RNN/All RSNNs) | | | 0.00001 / 0.1 | |
| **Model Configuration** | | | | |
| Hidden Neuron (RNN/All RSNNs) | | | 256 / 256 | |
| Token (Self-Attn H1/Self-Attn H2/Self-Attn H4) | | | 256 / 128 / 64 | |
| $\alpha$ (RNN/All RSNNs) | | | 0.01 / 0.01 | |
| $\alpha_{\text{noise}}$ (RNN/All RSNNs) | | | 0.5 / 0.5 | |
| $A_{\text{noise}}$ (RNN/All RSNNs) | | | 0.05 / 0.05 | |

## A.3 TRAINING SETUP

The parameter configuration in this study is shown in Table 1. When $\mathbf{P}_{\tau_d}$ is learnable in IP²-RSNN and RSNNs with randomized $\mathbf{m}$, each neuron is assigned two dendritic branches with sparse connectivity, following the setup described in Zheng et al. (2024).

## A.4 STRUCTURAL MODULARITY ANALYSIS

Structural modularity analysis involves decomposing a system into subsystems based on structural relationships. We assess modular organization using three key metrics: modularity, number of communities, and modular stationarity, and follow the configuration described in work (Gu et al., 2024).

Modularity $Q$ quantifies the modular organization of the functional connectivity network and is defined as:

$$Q = \frac{1}{2\mu} \sum_{ijlr} \left[ F_{ijl} - \gamma_l \frac{k_{il}k_{jl}}{2m_l}\delta_{lr} + \delta_{ij}C_{jlr} \right] \delta(g_{il}, g_{jr}), \tag{14}$$

here, $i$ and $j$ index neurons, $l$ and $r$ denote temporal layers (sliding windows over hidden-layer activity). $F_{ijl}$ is the adjacency (e.g., Pearson correlation) between nodes $i$ and $j$ in layer $l$. $\gamma_l$ controls resolution. $k_{il} = \sum_j F_{ijl}$ is the degree of node $i$ in layer $l$, and $m_l = \frac{1}{2}\sum_{ij} F_{ijl}$ is the total edge weight. $C_{jlr}$ denotes the inter-layer coupling strength linking node $j$ across layers $l$ and $r$, encouraging temporal continuity in community assignments. $\delta_{lr}$ and $\delta_{ij}$ are Kronecker delta functions, and $g_{il}$ denotes the community label of node $i$ in layer $l$. The modularity $Q$ is maximized to identify optimal communities across all time layers.

Community number quantifies the granularity of modular decomposition, with larger values indicating more fragmented network structures. For temporal layer $l$, the community number $C_l$ is defined as the number of distinct community labels assigned across all nodes:

$$C_l = |\{g_{il} \mid i = 1, \ldots, n\}|, \tag{15a}$$

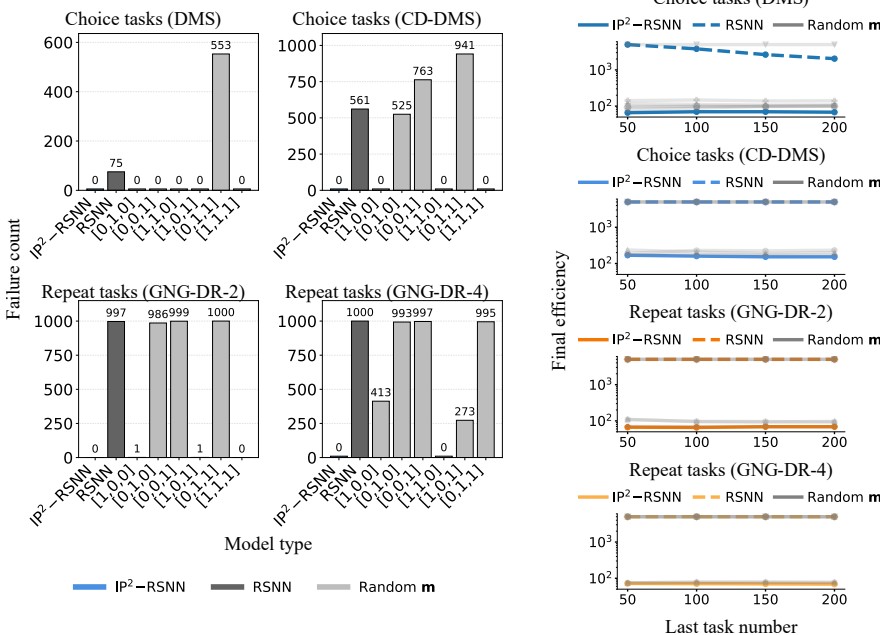

Figure S1: Comparison of L2L failure counts and final task efficiency among IP$^2$-RSNN, vanilla RSNNs, and RSNNs with randomized variations of **m** across four task families.

where $|\cdot|$ denotes the cardinality of the set, and $n$ is the total number of nodes. The overall community number $\bar{C}$ is then defined as the average across all $L$ temporal layers:

$$\bar{C} = \frac{1}{L} \sum_{l=1}^{L} C_l. \tag{15b}$$

Stationarity quantifies the temporal consistency of community structures. For a given community $c$, its stationarity $\zeta_c$ is defined as:

$$\zeta_c = \frac{1}{L_c - 1} \sum_{l=1}^{L_c - 1} \mathrm{corr}(\mathbf{v}_{c,l}, \mathbf{v}_{c,l+1}), \tag{16a}$$

where $\mathbf{v}_{c,l}$ is the membership indicator vector for community $c$ at time window $l$, and $L_c$ is the number of time windows in which community $c$ is detected. The overall stationarity $\bar{S}$ is computed as the average across all detected communities $\mathcal{C}$:

$$\bar{S} = \frac{1}{|\mathcal{C}|} \sum_{c \in \mathcal{C}} \zeta_c. \tag{16b}$$

Higher values of $\bar{S}$ indicate more stable modular structures over time.

### A.5 LLM USAGE

A large language model was used solely for grammar checking and minor phrasing suggestions for the manuscript text. The LLM did not contribute to research ideation, experimental design, model implementation, data generation, analysis, or results. All technical content, equations, figures, and code were created and verified by the authors.

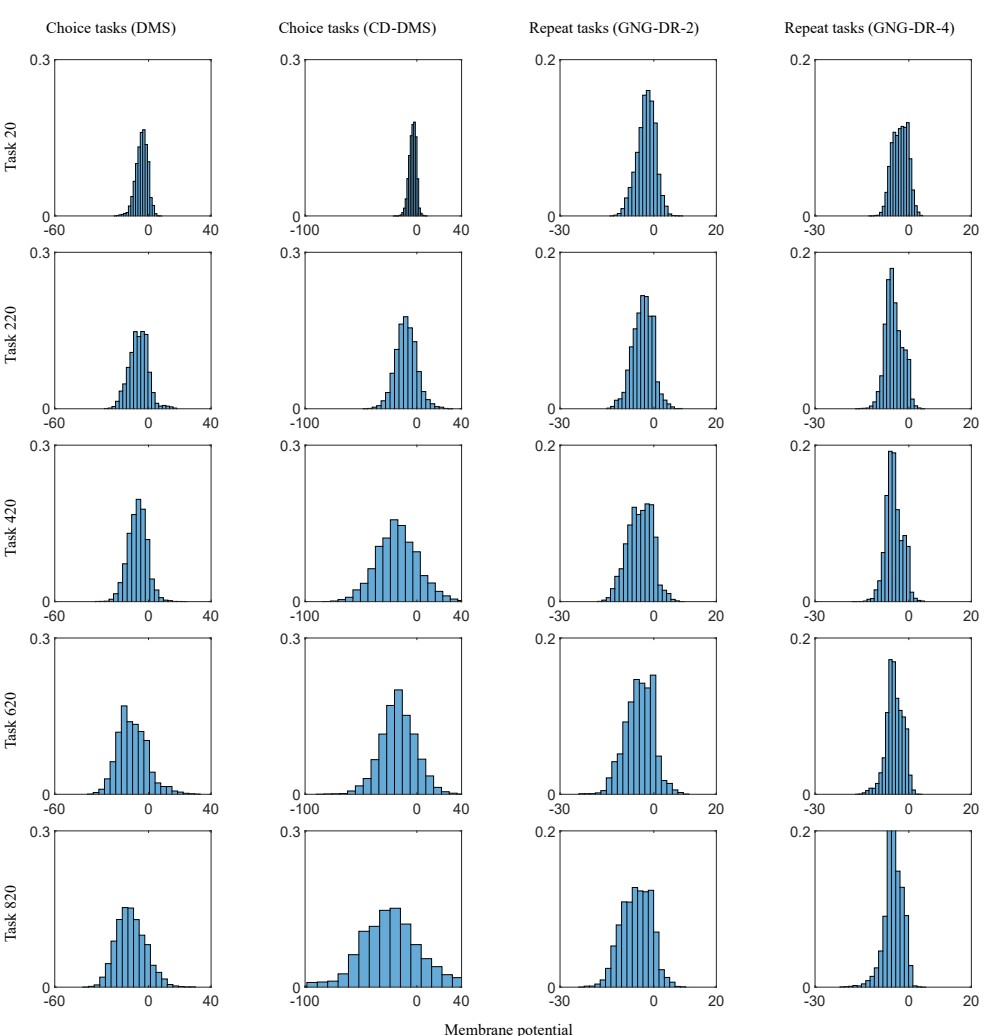

Figure S2: Evolution of IP²-RSNN membrane potentials during L2L across four task families.

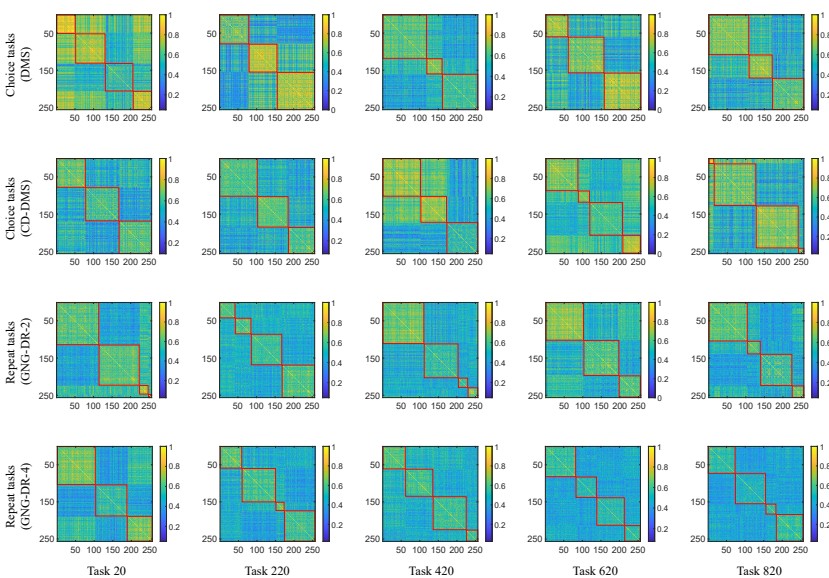

Figure S3: Evolution of the modular allegiance matrix in IP$^2$-RSNN during L2L across four task families.

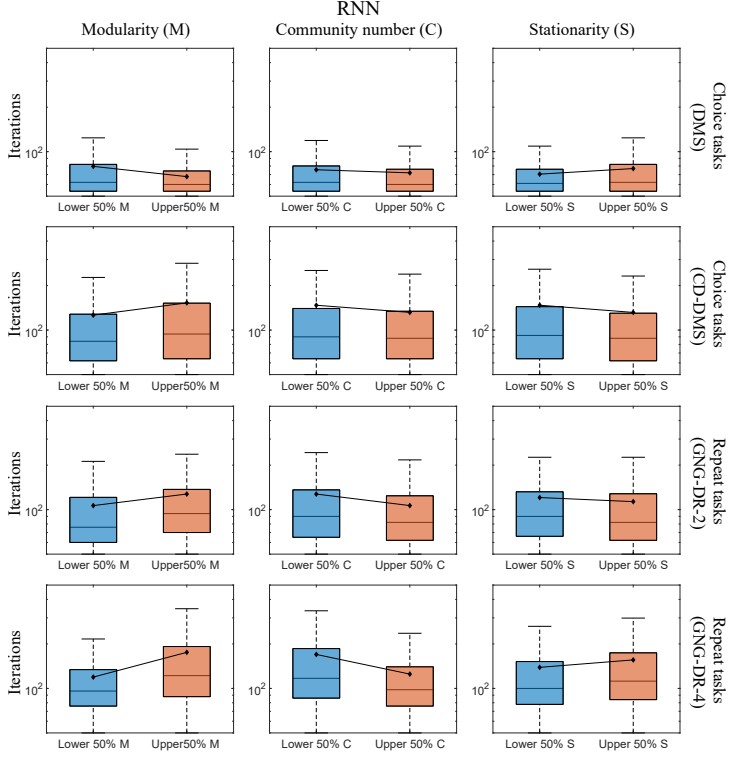

Figure S4: Relationships between three modularity metrics (modularity, number of communities, and stationarity) and adaptation speed in RNN across four task families.

