# OpenReview forum: "IP$^{2}$-RSNN: Bi-level Intrinsic Plasticity Enables Learning-to-learn in Recurrent Spiking Neural Networks"
_ICLR.cc/2026/Conference — ICLR 2026 Conference Withdrawn Submission_

### Official Review · Reviewer_TJ1Q · 2025-10-30

**Soundness:** 3
**Presentation:** 2
**Contribution:** 3
**Rating:** 4
**Confidence:** 2

**Summary:**

The authors present an SNN architecture with two-levels of meta-plasticity, aptly named IP^2-RSNN, which brings ideas from meta-learning and adaptive RNNs to SNNs.
The inner level of plasticity affects _learnable_ SNN parameters, optimizing them with respect to a task loss, membrane regularization, and weight penalties.
Meanwhile, the outer level of plasticity decides _which_ SNN parameters are learnable.
Unlike inner plasticity, the parameters involved in outer plasticity are static, in the sense that they should ideally only change when the nature of the current task changes, and should ideally be known before beginning task training and inference.

The authors evaluate their meta-plastic IP^2-RSNN architecture on four computational neuroscience tasks (two decision tasks and two recall tasks), and compare performance and efficiency against artificial neural networks (ANNs): RNNs and transformers.
They see that IP^2-RSNN models outperform ANNs in both task accuracy and data efficiency.
Additionally, they probe their IP^2-RSNN models to show that meta-plasticity allows task-specific adaptations that are not present in ANNs.

**Strengths:**

- The application of meta-plasticity to SNNs is novel.
- IP^2-RSNN is definitely promising, showing enhanced performance and efficiency over ANNs.

**Weaknesses:**

The largest issue with the authors' work is a lack of clarity.

Of particular note is the loss $\mathcal{L}$ used for meta-plasticity: it contains both task-specific ($\mathcal{L}\_{base}$) and task-agnostic terms, with the former being the principal driver of learning representations that are useful for a given task.
However, the authors do not clarify at what point the task-specific loss $\mathcal{L}\_{base}$ is hidden: do IP^2-RSNNs always have access to $\mathcal{L}\_{base}$ during inference?
Should readers interpret the results in Figure 3 as performance during training within an epoch of 1000 tasks, or are these results done entirely after training?
Moreover, are models trained on certain tasks, and then queried for inference on other tasks that they have not trained on, or do they train on all tasks, or does each task have a separately trained model?

The second major issue with the authors' work concerns the baseline models.

Details regarding the baseline ANN models (RNNs and transformers) are missing.
The largest Transformer models should be able to consistently complete the four tasks, although not necessarily as efficiently as the IP^2-RSNN. However, the Self-Attn H4 model struggles on the CD-DMS task, and it is unclear why: is the subpar performance because the model needs more data, or because of an inherent limitation in the model? This is where clarity in the training procedure would be useful.

Additionally, the claim that IP^2-RSNN outperforms ANN models may be misleading: ANN models are, to my knowledge, not allowed to update their parameters at inference time. Therefore, this claim should perhaps state that IP^2-RSNN outperforms fixed ANN models.
Substantiating that IP^2-RSNN outperforms ANN models should involve comparison with ANN models that are allowed to engage in meta-plasticity during inference.


My final issue with the author's work concerns their second set of results, which argue that adaptive parameters are important for adaptation.
I appreciate the authors' efforts in analyzing adaptation in adaptive SNNs, but it remains unclear to me what the essential takeaway is from this set of findings. One would expect that adaptive SNNs are better able to adapt during inference than non-adaptive networks like RNNs.
Perhaps a better comparison may be to other adaptive SNNs, such as LIF SNNs with simple spike-frequency adaptation mechanisms.

Additionally, there are no clear trends in Figures 6 and S4: to the naked eye, it would seem that RNN and transformer models also have variations in modularity metrics that correlate with efficiency. Is the reader supposed to conclude that the correlations in these models are different from those in IP^2-RSNN models, and thus inferior? If so, these figures should make that clear.

**Questions:**

See Weaknesses.

Minor suggestions:
- Figure 3: is CD-DMS Self Attn H1 missing?

---

### Official Review · Reviewer_9SMr · 2025-10-31

**Soundness:** 2
**Presentation:** 2
**Contribution:** 2
**Rating:** 4
**Confidence:** 4

**Summary:**

This paper proposes a new model called $IP^{2}$-RSNN, a recurrent spiking neural network that achieves "learning-to-learn" (progressively faster learning on similar tasks). It works by using a novel bi-level intrinsic plasticity: a slow meta-plasticity determines which intrinsic neuronal properties are learnable based on the task, while a fast plasticity fine-tunes those properties during training. The authors show this mechanism is essential for L2L and allows their model to outperform standard RNNs and self-attention models by enabling task-specific adaptations.

**Strengths:**

The paper proposes an interesting idea to control which intrinsic parameters are learnable based on task demands, rather than fine-tuning all of them. The paper uses a number of cognitive tasks to demonstrate the efficacy of such learnability control and analyzes how neuronal intrinsic parameters change over time. The paper also examines the relationship between network modularity and adaptation speed. Both the model and the analysis could be of potential interest to the computational neuroscience community.

**Weaknesses:**

1. The method section is unclear, with many details missing. I would also suggest adding pseudo-code to further clarify the outer-loop / inner-loop structure. See questions below.

2. While the paper describes the method as capturing “bi-level intrinsic plasticity,” the higher-level plasticity that controls the learnability of intrinsic parameters appears to be implemented simply by designating a one-hot mask for each task (Equation 6). However, it is unclear how this mask is quantitatively determined for a given task, and how such a meta-intrinsic plasticity function could be realized in either artificial or biological circuits. The biological motivation of the method is also unclear to me.

3. I am also unsure about the validity of the results, as tuning certain key hyperparameters could be critical for establishing the significance of the findings. See questions below.

**Questions:**

1. Is the model simultaneously trained on all 4 task classes, or do you use a different model for each task class? How are $P^{learnable}$ and $P^{fixed}$ initialized? When training on a new subtask, is $W$ retained or reinitialized? Could you add more details on how you generate data for each subtask?

2. When an intrinsic parameter is fixed, have you tried tuning its initial value? This could help determine if flexible, neuron-wise intrinsic parameters are necessary for the observed performance.

3. Could you provide more details about the baseline models? What was the simulation step size for the RNNs and self-attention models? Was any hyperparameter tuning performed for these baseline models? In particular, the learning rate can be a critical parameter for determining adaptation speed.

4. Could you also show a performance comparison in terms of average test loss versus the number of adaptation steps?

5. Could the authors provide more rationale from a neuroscience perspective for adapting intrinsic parameters as a way to accelerate learning? Is this considered a biologically plausible mechanism?

6. In Figure 2A, how would you explain why the adaptation iterations for the standard RSNN actually increase over time?

---

### Official Review · Reviewer_4L3x · 2025-10-31

**Soundness:** 2
**Presentation:** 2
**Contribution:** 3
**Rating:** 2
**Confidence:** 3

**Summary:**

The paper proposes a method for meta-learning or learning to learn (L2L) in spiking recurrent neural networks (RSNNs) through meta-optimisation and refinement of intrinsic neuronal properties, namely dendritic and somatic time constants and firing threshold (as opposed to optimising the weights of the network) in a sequential task-learning setting (sampling different input-output associations from same task family). The method comprises two nested optimisation loops: one outer slow meta-optimisation that selects which intrinsic variables are necessary to maintain as learnable over the task sequence; and an inner loop which fine-tunes the flexible intrinsic properties within each task on a faster time-scale.
The outer meta-optimisation is not properly described anywhere in the paper, but looks like an L1 optimisation on the vector $\mathbf{m}$.

The inner optimisation of the flexible intrinsic parameters is formulated in terms of gradient descent on a task-specific loss function that consists of five terms: [i] task performance, [ii] a homeostatic term for the recurrent activations that forces the squared activations towards some target values ,[iii-iv]  L2 penalties on the input, recurrent, and output weight matrices. (This point needs clarification because in the rest of the text the optimisation seems to be happening wrt the intrinsic neuronal parameters, however here the losses consider the weight matrices. Thus these three losses are either irrelevant since they don’t depend on the intrinsic parameters, or the authors indeed optimise also the weights, which means that there is a discrepancy wrt to the rest of the story of the paper and what they actually do).


The authors demonstrate the approach onto four task families (delayed match-to-sample (DMS), context-dependent DMS (CD-DMS), and two go/no-go delayed recall tasks (GNG-DR-2, GNG-DR-4)), each instantiated as 1000 sequential tasks in each task family, and they evaluate the “learning-to-learn” property in terms of **failure count** (number of tasks within a family on which the model fails to converge), **adaptation speed** (number of iterations required for each task to reach converge threshold), **final efficiency** (average adaptation speed over last x tasks) across tasks. They compare the performance of the proposed approach against a standard RNN and three transformers-based models with different number of attention heads (1, 2, and 4) and observe that the proposed method attains lower failure counts (Figure 3).
In Figure 2 they run 1000 task samples for each family and perform ablations by comparing two optimisation configurations (full bi-level IP and only inner IP). They show that the bi-level version performs better across all 4 task families and metrics.  Figure 4 takes a trained run, groups neurons by their learned intrinsic values, and lesions each group to characterise how much the next episode’s performance degrades, thereby arguing that the learned intrinsic heterogeneity is functionally specific.


I find the proposed approach of optimising only intrinsic neuronal parameters interesting, since it deviates form the usual weight optimisation training of recurrent networks, but it is yet unclear to me if this is really what the authors are doing. However, I have to admit that I was expecting the outer slower optimisation to adapt the synaptic weights, while the inner faster timescale modifies the intrinsic neuronal parameters.
Nevertheless I think the manuscript needs clarifications on several essential points, some I already mentioned and other I detail below.  The main points are that the optimisation is not described clearly, especially the meta-optimisation and how do the authors compute gradients through spikes; they need to describe better the training of a single task (each task index). Thus in my opinion the paper needs major updates for clarity, hopefully the authors can clarify during the rebuttal.

**Strengths:**

- The paper proposes an interesting approach for learning to learn that of optimising intrinsic neuronal parameters and selecting the subset of parameters that are important for each task type.
- The authors perform several ablation and lessioning experiments and observe that the modularity of networks evolved differently during training for different task types.

**Weaknesses:**

- The writing needs improvement. There is no clear description of the setting and the optimisation the authors perform.
- It is unclear if they optimise only the intrinsic parameters or both intrinsic parameters and weights.
- There is no description of the outer-loop (meta-) optimisation.
- See also questions and comments.

**Questions:**

# Comments

- In lines 42-43, the citations are a bit misleading. I was hoping to find papers that employ meta-learning studies to “providing insight into the mechanisms of L2L at the level of population dynamics”
- Lines 50-51: “In contrast, spiking neural networks (SNNs) offer greater biological realism, yet their application to L2L has been limited.” This is not really true, check the recent work from the Vogels lab [3].
- In Eq. 3, why do you need the Hadamart product, if eta is a scalar? If it is not a scalar, please make it boldface, and mention how do you adjust or select the learning rates for each dimensional component of the parameters.



# Questions

- How do you optimise the mask $\mathbf{m}$ that determines which intrinsic parameters are flexible or fixed. This seems like an L1 optimisation but I did not find anything neither in the main text, nor in the appendix. This is an essential part of the paper. Also how do you set the relative timescale of the two optimisations, and what is their influence on the resulting performance?
- How many trials do run for each single task (task index). I didn’t find anything in the text but I may have missed it.
- You have a spiking recurrent model, yet you read the output activity as a continuous time variable from the membrane potentials of the neurons [Eq. 13]. This modelling choice is a bit odd. Can you expand on the motivation behind this choice?
- How do you choose the lambda hyper parameters involved in the optimisation? Similarly how do you set the homeostatic targets $\sigma_h$?
- Can you expand on why do you need the optimisation terms that consider the three weight matrices? Isn’t the gradient of these terms wrt to the intrinsic parameters zero, or am I missing something here? In lines 151-152 you mention that you optimise also the model parameters. Does that mean that you adapt **both the weights and the intrinsic parameters** during training?  If yes, this is not clear in the rest of the text.
- Related to this regarding the optimisation, the work considers a spiking neuron model, yet there is no reference on how the gradient is computed.
- In lines 197-205 you seem to pre-specify the masks $\mathbf{m}$ for each task, which as I understand from the rest of the paper is the objective of the outer meta-optimisation. Can you expand on this?
- What exactly do you mean with task index, and how do you select the number 1000 for task index. Looking at Figure 3, it seems that the conclusions regarding the performance of the proposed framework against the other baselines would be different if you would consider task index up to 600-800. In particular, according to this figure the IP2-RSNN has zero failures across all four task types, while the RNN and the self-attention-H4 have zero task failures for the three task types, and a small number of failures for the context-dependent decision making task. Yet in the panels B in the second line it seems that for these two baseline models if you had selected task index 600, the performance would have been on par or better than the proposed approach. Can the authors explain this choice and discuss this matter?
- Also related to the last question, how do you explain the fact the fact that for the context dependent decision making task, the self-attention H2 has much less failures compared to the self-attention H4 model (2 vs 64).
- Also related to the last questions. The self-attention h1 model seems to not perform at all. So it is a rather bad choice for comparison. While it is nice to demonstrate that this model cannot work in the considered settings, it seems to obscure the results a bit. If we would exclude this model from the bar plots of figure 3A then the rest of the models perform almost equally well, with the exclusion of the CD-DMS, but the improvement looks less impressive, if we exclude the huge numbers that self-attention H1 model contributes. Just as a comment.
- In Figure 2B caption you mention “Roles of meta-intrinsic plasticity and intrinsic plasticity in IP2-RSNN disentangled by randomizing the learning mask m.” Can you explain this, because it is not obvious what is shown in the figure, and what argument you are trying to make.
- In figure 4 you show the evolution of the densities of the intrinsic parameters during the optimisation for the lesioned experiments. How is this evolution and final density different from an undamaged run. Also related, how does the final density depend on the initial conditions?


## Papers to be cited

I think the authors should cite the following paper from the Clopath lab that considers training RNNs by changing both synaptic weights and  neuronal excitability of neurons [1] (although not in a multi-task training setting), and this work [2] which considers multi-task training by optimising only task-specific biases for each neuron (which has similar flavour as optimising the “excitability” of each neuron for each task).

## Other comments

- between Eq. 1 and Eq. 2 I would suggest to add a phrase along the lines of: “or in component-wise form”.

---
## References

  [1] Delamare, Geoffroy, Douglas Feitosa Tomé, and Claudia Clopath. "Intrinsic neural excitability biases allocation and overlap of memory engrams." Journal of Neuroscience 44.21 (2024).

[2] Williams, E., Payeur, A., Ryoo, A. H. W., Jiralerspong, T., Perich, M. G., Mazzucato, L., & Lajoie, G. (2024). Expressivity of neural networks with random weights and learned biases. arXiv preprint arXiv:2407.00957.

[3] Confavreux, Basile, et al. "Memory by a thousand rules: Automated discovery of multi-type plasticity rules reveals variety & degeneracy at the heart of learning." bioRxiv (2025): 2025-05.

---

### Note · Authors · 2025-11-20

**Comment:**

We would like to withdraw this submission because the work requires substantial improvements and further validation. We plan to significantly refine the methodology and experimental analysis before resubmitting in the future.

**Withdrawal Confirmation:**

I have read and agree with the venue's withdrawal policy on behalf of myself and my co-authors.